# Circulating Extracellular Vesicles Impair Mesenchymal Stromal Cell Differentiation Favoring Adipogenic Rather than Osteogenic Differentiation in Adolescents with Obesity

**DOI:** 10.3390/ijms24010447

**Published:** 2022-12-27

**Authors:** Barbara Peruzzi, Enrica Urciuoli, Michela Mariani, Laura Chioma, Luigi Tomao, Ilaria Montano, Mattia Algeri, Rosa Luciano, Danilo Fintini, Melania Manco

**Affiliations:** 1Research Area for Multifactorial Diseases and Complex Phenotypes, Bambino Gesù Children’s Hospital, IRCCS, 00146 Rome, Italy; 2Unit of Endocrinology, Bambino Gesù Children’s Hospital, IRCCS, 00146 Rome, Italy; 3Department of Pediatric Hematology and Oncology, Bambino Gesù Children’s Hospital, IRCCS, 00146 Rome, Italy; 4Department of Laboratory Medicine, Bambino Gesù Children’s Hospital, IRCCS, 00146 Rome, Italy

**Keywords:** adolescent obesity, circulating extracellular vesicles, mesenand chymal stromal cells, osteogenic differentiation, physical exercise

## Abstract

Excess body weight has been considered beneficial to bone health because of its anabolic effect on bone formation; however, this results in a poor quality bone structure. In this context, we evaluated the involvement of circulating extracellular vesicles in the impairment of the bone phenotype associated with obesity. Circulating extracellular vesicles were collected from the plasma of participants with normal weight, as well as overweight and obese participants, quantified by flow cytometry analysis and used to treat mesenchymal stromal cells and osteoblasts to assess their effect on cell differentiation and activity. Children with obesity had the highest amount of circulating extracellular vesicles compared to controls. The treatment of mesenchymal stromal cells with extracellular vesicles from obese participants led to an adipogenic differentiation in comparison to vesicles from controls. Mature osteoblasts treated with extracellular vesicles from obese participants showed a reduction in differentiation markers in comparison to controls. Children with obesity who regularly performed physical exercise had a lower circulating extracellular vesicle amount in comparison to those with a sedentary lifestyle. This pilot study demonstrates how the high amount of circulating extracellular vesicles in children with obesity affects the bone phenotype and that physical activity can partially rescue this phenotype.

## 1. Introduction

In recent decades, the obesity epidemic in children and adolescents has become one of the most relevant health issues in Western countries. Overweight and obesity in childhood have been related to many co-morbid conditions such as metabolic, cardiovascular, orthopedic, neurological, hepatic, pulmonary and renal disorders [1].

Extracellular vesicles (EVs) are membrane-derived nanoscale particles secreted by virtually any cell in the extracellular space [2]. EVs are classified into microvesicles, exosomes and apoptotic bodies on the basis of their biogenesis, release mechanism and size, and have been found to convey proteins, lipids and nucleic acids (DNA, RNAs and miRNAs), as well as metabolites as cargo material [3,4]. By transporting bioactive molecules, EVs can target and condition neighboring cells within the microenvironment in a autocrine/paracrine manner, as well as enter the biologic fluids to reach distant recipient cells inducing transient or persistent phenotypic changes [5].

Although EVs and their release in the bloodstream are now accepted as a physiological phenomenon involved in intercellular/inter organ communication, they are also crucial mediators of many pathological conditions, from viral infections [6,7] to cardiac and neurodegenerative diseases [8,9], as well as cancers [10] and many other pathologies [11].

In this context, recent studies have indicated that EVs play a critical role in the pathogenesis of obesity [12], highlighting their relevant roles in exacerbating obesity-related chronic inflammation and the consequent immune system deregulation [13] and metabolic complications [14]. Here, we investigated the effects of circulating EVs in children with obesity on bone cell differentiation and activity to shed light on the controversial bone phenotype observed in adolescent obesity. Indeed, excess body weight due to obesity has traditionally been considered beneficial to bone health because of a well-established anabolic effect of the mechanical loading conferred by body weight on bone formation [15]. However, there is debate that excess weight is detrimental to bone, resulting in an increased fracture risk [16,17]. This obesity-related impairment of bone quality is at least in part due to an increased bone marrow adiposity [17], a condition in which mesenchymal stromal cells are driven towards adipogenesis rather than osteogenic differentiation [18,19]. Indeed, a higher risk of fracture in obese people is observed despite the normal or even increased bone mineral density [20], suggesting local and/or systemic factors compromise bone morpho-structural properties and likely reduce bone turnover by favoring bone marrow fat depots.

Therefore, our working hypothesis was that circulating extracellular vesicles in children with obesity could be a pivotal factor impairing mesenchymal stromal cell differentiation and osteoblast activity. Furthermore, we hypothesize that increased physical activity favors bone mineral accrual in growing children by reducing circulating EVs regardless of obesity status.

## 2. Results

Thirty-two out of eighty-two (39%) enrolled participants were normal weight with no history of excess weight; twenty-eight (34%) were overweight and twenty-two were obese (27%). Their anthropometrics and biochemistry are shown in Table 1.

### 2.1. Participants with Overweight/Obesity Had a Higher Amount of Circulating Evs Than Normal-Weight Peers

In order to quantify the amount of circulating EVs in the bloodstream of participants with normal weight (NW), overweight (Ow) and obesity (Ob), we collected EVs from plasma samples and stained them with CFSE dye to point out green cytoplasm, thereby achieving CFSE^+^ EVs. We used beads with a known size of 1.34 μm to create a gate (size <1 μm) in which we expected to find the EVs (Appendix A). Moreover, to compare all the FACS analysis of EVs, we added an equal amount of CountBright™ absolute counting beads to each sample (Appendix A) and used the fluorescent signal of CFSE^+^ EVs to distinguish these nanoparticles from background noise in each sample (Appendix A). Next, we analyzed the amount of CFSE^+^ EVs in participants’ samples, finding a statistically significant difference among groups (*p* = 0.030 by one-way ANOVA test) in the higher amount of circulating EVs in both overweight and obese samples in comparison to those derived from normal weight peers (Figure 1). This result suggested a linear correlation between the amount of circulating EVs and the BMI, as confirmed by the comparison between children with a BMI SDS ≤ 1.02 (normal weight children) and children with a BMI SDS > 1.02 (overweight and obese patients) (Appendix A). Univariate analysis found age (coeff. 0.28; *p* = 0.02), BMI z-score (coeff.0.25; *p* = 0.04), FI (coeff. 0.24; *p* = 0.04) and HOMA-IR (coeff. 0.255, *p* = 0.04) predicted the number of EVs.

### 2.2. Uptake of Circulating EVs by Mesenchymal Stromal Cells

We treated mesenchymal stromal cells (MSCs), as precursors of the bone-forming osteoblastic lineage, with CFSE ^+^ EVs collected from the plasma of participants with normal weight, overweight and obesity. The resulting uptake of EVs was assessed in a time course experiment by fluorescent microscopy analysis (Figure 2A), demonstrating that EVs derived from different patients had the ability to merge into target cells with a comparable efficiency, as indicated by the assessment of fusion percentage of CFSE ^+^ MSCs (data not shown). At the same time, the quantification of the CFSE mean fluorescence intensity (MFI) in MSCs, as an index of CFSE ^+^ EV fusion in target cells, demonstrated a significantly higher uptake in cells treated with obese EVs at early time points (0.5 and 1 h) in comparison to EVs from the other groups (Figure 2B and Appendix A). This finding revealed an increased affinity and/or tropism of obese EVs to mesenchymal stromal cells, in comparison to normal weight and overweight participants.

### 2.3. Treatment of EV from Obese Patients on Mesenchymal Stromal Cells Affects Both Osteogenic and Adipogenic Differentiation

To elucidate the effects induced by the treatment with normal weight, overweight and obese EVs, MSCs were treated with circulating EVs and assessed for adipogenic and osteogenic differentiation. We investigated whether the EV treatment alone, in a proliferating medium, was able to trigger/impair adipogenic and/or osteogenic differentiation, in comparison to canonical MSC differentiation achieved using adipogenic and osteogenic differentiation media. Indeed, normal weight, overweight and obese EVs were added to the proliferating medium (PM) and refreshed every 3 days until complete differentiation was achieved in the canonical conditions (21 days of treatment with differentiation media, DM). Beside the canonical osteoblastic differentiation observed in MSCs treated with canonical osteogenic differentiation medium (positive control condition), we observed a basal, albeit minimal, osteoblast differentiation in the negative control condition (MSCs in proliferating medium without EV treatment), as demonstrated by alkaline phosphatase (ALP) staining (Figure 3A) and quantification (Figure 3B).

The treatment of MSCs with obese EVs further reduced the baseline osteoblast differentiation of the negative control condition and significantly impaired the induction of differentiation achieved by the treatment with normal weight EVs (Figure 3A,B). Notably, the treatment with normal weight, overweight and obese EVs caused a gradual decreasing effect on osteogenic differentiation, inducing a differentiation in the normal weight EVs, no significant effect on the overweight EVs and a reducing effect on the obese EVs, in comparison to the negative control. With regard to the adipogenic differentiation of MSCs achieved by canonical treatment with adipogenic differentiation medium (positive control condition) and assessed by oil red staining (Figure 3A) and quantification (Figure 3C), we observed an induced adipogenic differentiation in MSCs solely treated with overweight and obese EVs (Figure 3A,B), with a gradual increase in differentiation from normal weight EVs to obese EVs. These findings on the effects mediated by circulating EVs on MSC osteogenic and adipogenic differentiation were also confirmed by quantitative RT-PCR by the assessment of Runx2 (Figure 3D) and PPAR-γ (Figure 3E) gene expression, the master genes of osteogenic and adipogenic differentiation, respectively.

### 2.4. Uptake of Circulating EVs by Mature Osteoblasts

We then investigated the effects of circulating EVs collected from normal weight, overweight and obese children on mature osteoblasts, the bone-forming cells. With regard to the timing and the rate of fusion on target cells, no significant differences were found among patients’ EV uptake, and a different timing of fusion was detected (higher fusion at late time points) (Figure 4A,B and Appendix A) in comparison to results achieved with mesenchymal stromal cells. This result suggested that, although circulating EVs are still able to fuse into mature osteoblasts, the prominent fusion efficiency of obese EVs on MSCs appeared to be attenuated.

### 2.5. Obese EV Treatment Impairs Osteoblast Differentiation

To further investigate the effects of circulating EVs on mature osteoblasts, these cells were treated with normal weight, overweight and obese participants’ EVs for 72 h and then assessed for any modulation of differentiation. Figure 5A shows a reduction in ALP staining following the treatment of mature osteoblasts with EVs from overweight and obese participants, in comparison to the treatment with normal weight participant EVs. This observation was confirmed by ALP quantification, in terms of both intensity (Figure 5B) and percentage (Figure 5C) of staining. Moreover, quantitative RT-PCR on the expression of ALP (Figure 5D) and Runx-2 genes (Figure 5E) confirmed the impairment of osteoblast differentiation induced by the treatment with obese and overweight participants’ EVs.

### 2.6. Physical Activity Counteracts the Increase in Circulating EVs in Obese Children

To further investigate the effects of circulating EVs on the systemic phenotype associated with obesity in youth, we hypothesized the involvement of exercise and physical activity, as based on evidence in the literature. Since exercise can influence EV release [21], we reevaluated the amount of circulating EVs in participants with normal weight, overweight and obesity in relationship to declared daily lifestyle. Indeed, a 2-way ANOVA test demonstrated that the release of EVs was significantly influenced by both the BMI and sport activity, with the amount of circulating EVs in participants with obesity practicing physical exercise comparable to those circulating in normal weight participants (Figure 6A).

Indeed, physical exercise is also responsible for increasing the systemic level of osteocalcin [22,23], a hormone known to regulate glucose homeostasis [24] and insulin sensitivity [25,26]. Therefore, we compared the metabolic parameters and the number of vesicles of the participants who had regularly practiced endurance sport activities at least three hours a week in the six months preceding the study with those of participants who had not practiced any sport or physical activity. We found significantly reduced HOMA-IR and fasting insulin (Figure 6B) and post load insulin (Figure 6C) in participants who regularly practiced endurance PA vs. those who did not across the three groups of participants with NW, Ow and Ob. No difference was found in fasting and post load glucose concentration among groups (Appendix A). Two-way ANOVA post hoc analysis revealed that both the “weight” factor and the “physical activity” factor are significantly involved in regulating assessed metabolic indices. Additionally, plasma insulin levels post glucose load were significantly reduced in children with obesity practicing sport activity (Figure 6C), while post load glucose levels were not modulated (Appendix A), suggesting that physical exercise is responsible for modulating systemic metabolic parameters in a specific manner.

## 3. Discussion

Our findings demonstrate an increased number of circulating EVs in overweight and obese adolescents as compared to peers with normal weight, and that obese participants’ EVs are responsible for impairing MSC in vitro differentiation by favoring adipogenesis at the expense of osteoblastogenesis.

Higher amounts of circulating EVs have been found in the bloodstream of obese adult patients in comparison to normal weight control subjects [27]. Here, we demonstrated that the relationship between circulating EV amount and BMI is retained also in obese adolescents. Indeed, circulating EVs were associated with age, body weight and BMI z score, but also with fasting insulin and HOMA-IR. A multivariate analysis failed to rule out whether the association between EVs and HOMA-IR is independent of the excess adiposity.

In view of this, the number of circulating EVs may represent a metabolic biomarker in obesity given its direct association to adipocyte number and hypertrophy, and to obesity-related long-term metabolic disorders, such as diabetes and kidney failure [28].

Interestingly, we described an impairment in mesenchymal stromal cell differentiation and osteoblast function following treatment with EVs from obese participants, which could be related to the bone phenotype observed in obese children [17,29]. Notably, the timing and the efficiency of EV uptake was significantly different among the groups of collected EVs (children with normal weight, overweight and obesity) and between recipient cells (mesenchymal stromal cells vs. mature osteoblasts). Indeed, we demonstrated that EVs from obese participants had a higher efficiency of fusion into mesenchymal stromal cells, in comparison to those from normal weight and overweight participants. This finding could depend on the higher number of vesicles collected from children with obesity in comparison to the other groups of patients. Indeed, we found an overall reduced uptake and no differences among EV groups in cell targeting when different recipient cells (mature osteoblasts) were used, suggesting that EVs from obese participants convey some specific surface proteins in their cargo, such as tetraspanins protein and integrins, which are responsible for increasing specific cell natural targeting for mesenchymal stromal cells, a phenomenon already described in the literature [30,31,32]. The high affinity of EVs from obese participants to MSC targeting results in a functional effect, i.e., an impairment of their lineage differentiation favoring adipogenesis and hindering osteogenesis. This finding provides new insights into the characterization of the bone phenotype and the bone marrow adiposity observed in obese children [17,33]. Although several papers in the literature have shown EV-mediated effects on MSC differentiation [34,35], this work demonstrates for the first time a key pivotal role of circulating EVs in the establishment of the metabolic and bone alteration in pediatric obesity.

Indeed, we found that performing physical activity affects metabolic parameters in youths with obesity, restoring normal values of HOMA-IR index and fasting insulin, as well as reducing the amount of circulating EVs to a level comparable to normal weight controls. According to the literature, physical exercise is responsible for rapidly increasing the amount of circulating EVs which, in healthy conditions, is mainly skeletal-muscle derived [21,36,37]. We found a reduction in circulating EVs following physical activity, which is in contrast to data in the literature. Indeed, exercise-mediated effects on the release of EVs have been described as tissue-, sex-, age- and BMI-dependent [38,39,40]. Our results showed that physical activity is statistically related to a decreased amount of circulating EVs and to a recovery of metabolic parameters in youths with obesity. These findings confirm that EVs play a relevant role in the metabolic alteration associated with pediatric obesity and suggest that EVs are mediators of the physical activity-related beneficial effects on bone phenotype and diabetes risk.

To explain the EV-mediated effects on bone cells, an in-depth characterization of the molecules contained within the EVs from obese participants needs to be undertaken in future work. Regardless, among the candidate factors conveyed as cargo in EVs from obese participants, osteocalcin may be involved in modulating the metabolic parameters and the bone cells, given its involvement in regulating physiologic processes such as insulin and glucose homeostasis [26,41,42] and the correlation between osteocalcin production and physical exercise [23]. In addition to osteocalcin, it is also conceivable that Interleukin(IL)-6 is conveyed as a cargo within the EVs from obese participants, based on data in the literature showing that IL-6 serum level are elevated in children with obesity [43] and are elevated following acute exercise [44]. Interestingly, high IL-6 levels have been described to impair bone turnover in a growing skeletal system by decreasing osteoblast activity [45,46], in a very similar manner to our results.

Despite the novelty and the relevance of our results, this study has some limitations. The first of which is the sample size and the methodological criticalities in working with circulating extracellular vesicles. Secondly, a thorough characterization of the bioactive content of these vesicles, in terms of proteins, lipids and nucleic acids, will greatly validate the translational relevance of this study.

## 4. Materials and Methods

### 4.1. Study Sample

Eighty-two adolescents (aged between 10 and 17 years) were consecutively enrolled at the Endocrinology Unit of the Bambino Gesù Children’s Hospital between May 2020 and 2021. They belonged to the “Bambino” meta-cohort [47,48]. The ‘Bambino’ meta-cohort included normal weight children and adolescents with excess weight who participated in studies on obesity run between 2006 and 2016 at the Bambino Gesù [49,50,51,52,53]. The inclusion criterion was a stable body weight in the 6 months prior to the study, the exclusion criteria were surgical treatment of obesity, presence of any systemic and endocrine disease and use of any medication, including contraception and alcohol or recreational drugs.

### 4.2. Lifestyle Habits and Physical Activity

Information on socio-economic status and lifestyle habits was collected by questionnaire. As for physical activity, an adapted version of the Physical Activity Questionnaire for Adolescents (PAQ-A) was conducted [54,55]. The questionnaire is a 7-day recall questionnaire that measures general moderate to vigorous physical activity levels during a school year. It asks about the time spent in physical activities in four different settings (domains): (A–B) school-related physical activity, including activity during physical education classes and breaks; (C) housework, house maintenance and gardening; (D) transportation; and (E) recreation, sport and leisure-time physical activity. The questionnaire comprises nine or eight items (PAQ-C includes an additional item on recess) and collects information on participation in different types of activities and sports (activity checklist), effort during physical activity and activity during lunch, after school, evening and at the weekend over the past 7 days. Each item was scored between 1 (low PA) and 5 (very high PA) and the average score denotes the PAQ score. We defined those scoring ≥2.6 as physically active, which was the median value reported in our sample, and those scoring below 2.6 as low active.

### 4.3. Anthropometric Measurements and Biochemical Assays

Body mass index (BMI) and sex- and age-specific standard deviation scores (SDS) of BMI were calculated [56]. Normal weight was defined as BMI z-score (SDS) < 1.02; overweight as BMI z-score > 1.04; and obesity as BMI z-score > 1.64 [57,58].

Overweight and obese participants with overweight and obesity underwent a standard OGTT (1.75 g/kg body weight up to a maximum of 75 g) with flavored glucose (Glucosio Sclavo Diagnostics, 75 g/150 mL). Fasting glucose (FG) and post load values were measured by the glucose oxidase technique (Cobas Integra, Roche Diagnostics S.p.A., Italy) and fasting insulin (FI) and post load values by the chemiluminescent immunoassay method (ADVIA Centaur analyzer; Bayer Diagnostics, Germany). We used different units for FG and FI (mg/dl and µlU/mL, respectively) for the calculation of indices. In all participants, indices of insulin sensitivity and beta cell function, namely HOMA-IR and HOMA-B, were calculated using the HOMA calculator provided by the University of Oxford (https://www.dtu.ox.ac.uk/homacalculator/ (accessed on 24 November 2021)). In overweight and obese participants, the whole-body Insulin Sensitivity Index (ISI) was also calculated [59]. All participants were asked to refrain from intensive physical activity in the 3 days prior to the study.

### 4.4. Isolation and Staining of Extracellular Vesicles from Patients’ Plasma

Extracellular vesicles (EVs) were isolated from 200 µL of plasma sample using 60 µL of ExoQuick^®^ Exosome Isolation Reagent (SBI, System Biosciences, Palo Alto, CA, USA) according to the manufacturer’s instructions. EV pellets were suspended in sterile PBS. For FACS analysis and uptake experiments, isolated EVs were stained with 5 uM CFSE dye (ThermoFisher Scientific, Waltham, MA, USA) for 30 min at 37 °C. For EV treatment of cells (uptake and functional experiments), EVs from each group were pooled and added in a volume of 3% of the cell culture medium volume. We chose to fix the volume, rather than the number, of each EV pool to recapitulate the different amounts of circulating EVs found in patients’ samples. Given the paucity of plasma, in terms of volume, in each patient, it was not possible to perform any further analyses or characterization of circulating extracellular vesicles.

### 4.5. Flow Cytometry Analysis

To visualize and quantify CFSE^+^ EVs, we used the protocol described in work by Urciuoli et al. [60]. Briefly, nanobeads (Spherotech, Inc., Lake Forest, IL, USA) with a known size of 1.3 μm were used to create the gate in which to visualize EVs (<1 µm). Moreover, we used CountBright™ absolute counting beads ( ThermoFisher Scientific, Waltham, MA, USA) to perform a reproducible evaluation by analyzing the volume containing 1.000 beads (5 μL) for each sample. CFSE ^+^ EVs were analyzed with FACS LSR FORTESSA X–20 (BD PharMigen, BD Biosciences, Franklin Lakes, NJ, USA) and data were calculated using the FACS DIVA 8.0 software (BD PharMigen, BD Biosciences, Franklin Lakes, NJ, USA).

### 4.6. Cell Cultures

Primary human osteoblasts (HOB) (catalog #4610) were purchased from ScienCell™ (Carlsbad, CA, USA). HOB were grown in DMEM (Euroclone Spa, Pero, Italy) supplemented with 10% fetal bovine serum (FBS) and 100 units/mL penicillin/streptomycin (Euroclone Spa, Pero, Italy) and maintained at 37 °C in 5% CO2. MSCs were isolated and expanded ex vivo from BM aspirates of healthy donors (HDs) who donated BM for hematopoietic cell transplantation at Bambino Gesù Children’s Hospital and were used as controls.

To isolate the MSCs, a density gradient centrifugation (Ficoll 1077 g/mL; Lympholyte, Cedarlane Laboratories Ltd., The Hague, The Netherlands) was performed to collect bone marrow-mononuclear cells (BM-MNCs). BM-MNCs were then washed twice in saline phosphate buffer (PBS, Euroclone Spa, Pero, Italy) and seeded at a density of 160 × 10^3^/cm^2^ in DMEM low glucose (Euroclone Spa, Pero, Italy), 10% FBS (Gibco, ThermoFisher Scientific, Waltham, MA, USA), 2 mmol/L-glutamine and 100 g/mL penicillin-streptomycin (Euroclone Spa, Pero, Italy). After at least 36 h, non-adherent cells were removed and the culture medium was replaced twice a week. MSCs were then harvested after reaching ≥80% confluence with a Trypsin solution (Euroclone Spa, Pero, Italy) and then transferred to a new flask at a concentration of 4 × 10^3^ cells/cm^2^ for the subsequent passages. All MSCs obtained were confirmed to be negative for mycoplasma by routine testing performed once every month.

### 4.7. EV Uptake

MSCs and HOBs were seeded on plastic-chambered glass microscope slides (BD Falcon, Franklin Lakes, NJ, USA) at a density of 20 × 10^3^ and 10 × 10^3^, respectively, and treated with 3% of CFSE^+^ EVs for 0.5, 1, 3 and 6 h to evaluate EV fusion into target cells and the percentage of MSCs and HOBs positive to CFSE after EV uptake. The fluorescence intensities of CFSE were acquired by confocal microscopy.

### 4.8. Confocal Microscopy Analysis

Confocal microscopy imaging was performed on a Leica TCS-SP8X laser-scanning confocal microscope (Leica Mycrosystem, Mannheim, Germany) equipped with tunable white light laser source and a 405 nm diode laser. Sequential confocal images were acquired using a HCPLAPO 40x oil-immersion objective with a 1024 × 1024 format, a scan speed of 400 Hz and z-step size of 0.25 μm. The fluorescence intensities of CFSE and Hoechst were calculated using ImageJ software (NIH, Bethesda, MD, downloadable at http://rsbweb.nih.gov/ij/download.html, accessed on 24 November 2021) from cytometric measurements in 5 digital images randomly selected and acquired for each cell sample with a 512 × 512 format and a scan speed of 400 Hz.

### 4.9. MSCs Treatment

MSCs were seeded with αMEM (Euroclone Spa, Pero, Italy) and 10% ultracentrifuged FBS (in order to remove endogenous FBS EVs) and supplemented with 3% isolated EV. Media were changed every 3 days, and after 3 weeks of culture, canonical differentiation staining and gene expression analysis were performed. MSCs treated with osteogenic and adipogenic media were used as the positive control. To assess the osteogenic differentiation, alkaline phosphatase (ALP) staining was evaluated by an Alkaline Phosphatase Activity Kit No. 86C (Sigma-Aldrich, St Louis, MO, USA), while the adipogenic differentiation was evaluated through the staining of fat droplets with oil red O (Sigma-Aldrich, St Louis, MO, USA).

### 4.10. HOBs Treatment

HOBs were seeded with DMEM (Euroclone Spa, Pero, Italy) and 10% ultracentrifuged FBS (in order to remove endogenous FBS EVs), and supplemented with 3% isolated EV for 72 h. The osteogenic differentiation was assessed by an Alkaline Phosphatase Activity Kit No. 86C (Sigma-Aldrich, St Louis, MO, USA), while the gene expression analysis was evaluated by quantitative RT-PCR.

### 4.11. RT-qPCR

Total RNA was extracted from cultured cells using the standard Trizol procedure. Each RNA sample was quantified by NanoDrop 2000 (ThermoFisher Scientific, Waltham, MA, USA). Two micrograms of RNA were reverse transcribed using the SuperScript™ II Reverse Transcriptase (ThermoFisher Scientific, Waltham, MA, USA) to generate cDNA. The PCR reaction was carried out with Power SYBR Green dye chemistry (Applied Biosystems by ThermoFisher Scientific, Waltham, MA, USA) using the QuantStudio™ 7 Pro Real-Time PCR System (Applied Biosystems, Foster City, CA, USA). Results were normalized to GAPDH levels using the 2−ΔΔCt method. Primer pairs 5′-3′: Human GAPDH reverse: AGGGGTCTACATGGCAACTG; human GAPDH forward: CGACCACTTTGTCAAGCTCA. Human ALP reverse: CCACCAAATGTGAAGACGTG; human ALP forward: GGACATGCAGTACGAGCTGA. Human RUNX2 reverse: TATGGAGTGCTGCTGGTCTG; human RUNX2 forward: TTACTTACACCCCGCCAGTC. Human PPAR-γ reverse: TGGGCGGTTGATTTGTCTGT; human PPARγ forward: CTTGCAGTGGGGATGTCTCA.

### 4.12. Statistical Analysis

The molecular data were presented as box plot graphs or means ± standard deviation graphs. All the statistical analyses were performed on at least three independently replicated experiments. Comparisons among groups were performed by using one-way ANOVA-Multiple comparisons, while the time course experiments of EV fusion into MSCs and HOBs were analyzed by using two-way ANOVA-Multiple comparisons. Univariate analyses were run to identify clinical variables predicting the number of EVs. All analyses were performed using GraphPad Prism 6.0. (GraphPad software Inc., La Jolla, CA, USA). A *p*-value < 0.05 was considered significant.

## Figures and Tables

**Figure 1 ijms-24-00447-f001:**
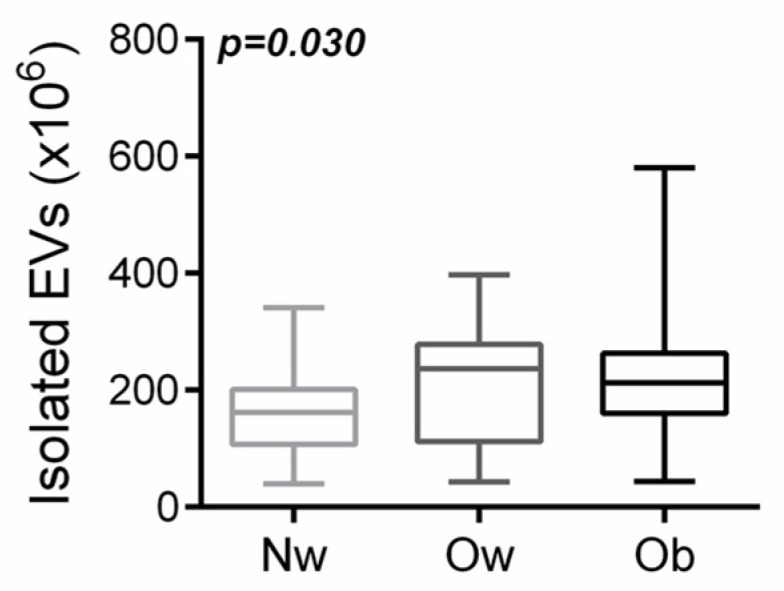
Quantification of circulating EVs by FACS analysis in participants with normal weight (Nw), overweight (Ow) and obesity (Ob).

**Figure 2 ijms-24-00447-f002:**
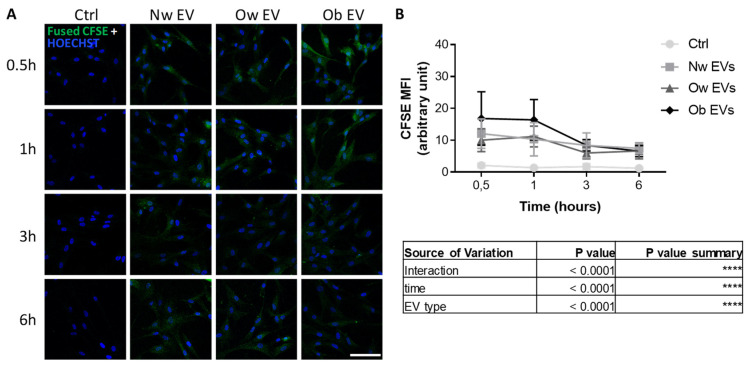
(**A**) Confocal microscopy analysis of MSC treated for 0.5, 1, 3 and 6 h with CFSE^+^ EVs from participants with normal weight (Nw), overweight (Ow) and obesity. Pictures show the resultant recipient cells as proof of EV uptake. Nuclei were stained with Hoechst (blue). Scale bar: 100 μm. (**B**) Mean fluorescence intensity (MFI) quantification of CFSE ^+^ MSCs following the treatment with CFSE^+^ EVs. The table below shows the results from ANOVA 2-way analysis.

**Figure 3 ijms-24-00447-f003:**
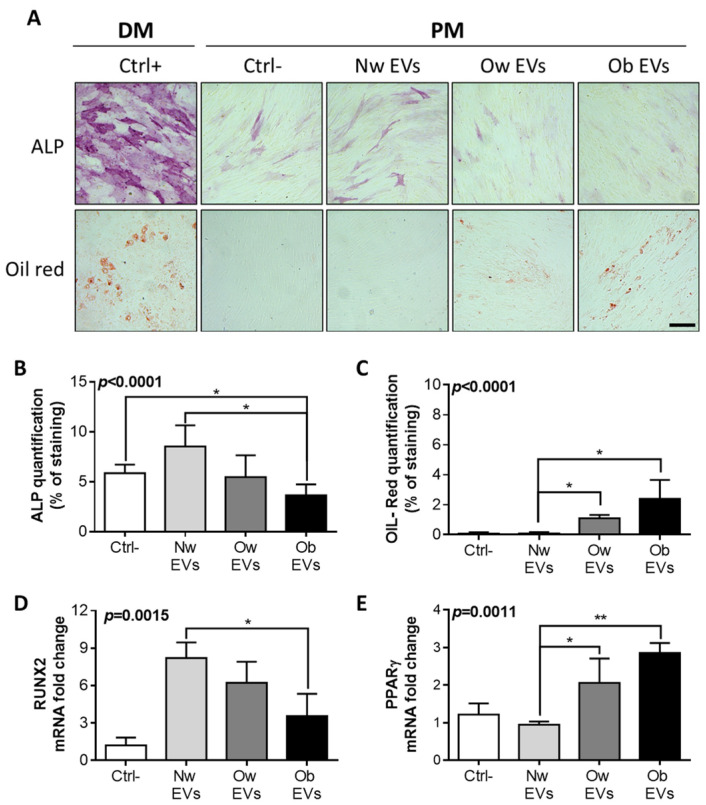
(**A**) MSC were treated for 21 days with canonical differentiation medium (DM) to achieve a complete osteogenic (upper panel) and adipogenic (lower panel) differentiation evaluated by alkaline phosphatase (ALP) and oil red staining, respectively. This condition served as positive control (Ctrl+) of the experiment. In parallel, MSC were cultured for 21 days in a proliferation medium (PM) and treated every 3 days with EVs collected from children with normal weight (NW EVs), overweight (Ow EVs) and obesity (Ob EVs), and compared to control condition (Ctrl-) with no EV treatment. These samples were also stained for ALP and oil red to assess osteogenic (upper panel) and adipogenic (lower panel) differentiation, respectively. Scale bar: 200 μm. Quantification of the percentage of ALP (**B**) and oil red (**C**) staining in MSCs treated with EVs in PM for 21 days. RT-PCR for the expression of osteoblastic master-gene Runx2 (**D**) and the adipogenic master-gene PPAR-γ (**E**) in MSCs treated with NW, Ow and Ob EVs for 21 days, normalized versus the housekeeping GAPDH. * *p* < 0.05; ** *p* < 0.01.

**Figure 4 ijms-24-00447-f004:**
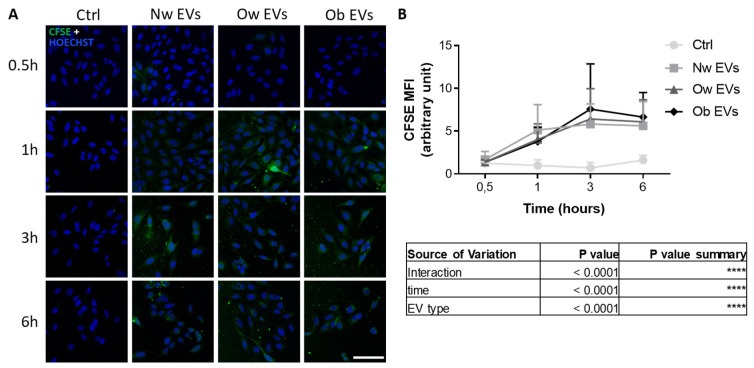
(**A**) Confocal microscopy analysis of mature osteoblasts treated for 0.5, 1, 3 and 6 h with CFSE^+^ EVs from participants with normal weight (NW), overweight (Ow) and obesity. Pictures show the resultant recipient cells as proof of EV uptake. Nuclei were stained with Hoechst (blue). Scale bar: 100 μm. (**B**) Mean fluorescence intensity (MFI) quantification of CFSE^+^ osteoblasts following the treatment with CFSE^+^ EVs. The table below shows the results from ANOVA 2-way analysis.

**Figure 5 ijms-24-00447-f005:**
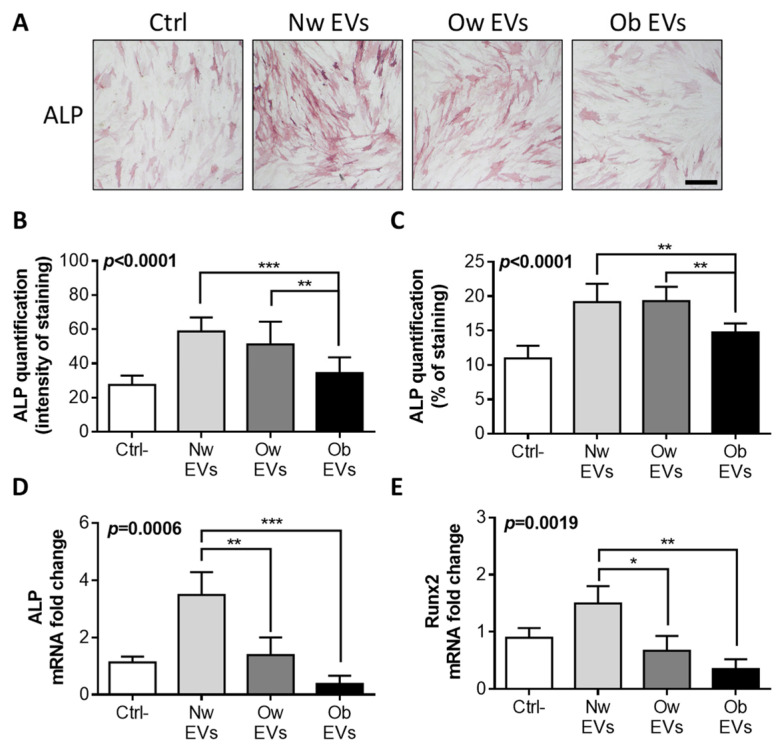
(**A**) Human primary mature osteoblasts were treated for 3 days with EVs collected from children with normal weight (NW EVs), overweight (Ow EVs) and obesity (Ob EVs), and compared to the control condition (Ctrl-) with no EV treatment. At the end of the experiment, mature osteoblasts were stained for ALP to assess any modulation in the osteogenic differentiation. Scale bar: 200 μm. Quantification of the ALP intensity of staining (**B**) and percentage of staining (**C**) in osteoblasts treated with EVs for 3 days. RT-PCR for the expression of osteoblastic differentiation marker ALP (**D**) and the differentiation master-gene Runx-2 (**E**) in osteoblasts treated with NW, Ow and Ob EVs for 3 days, normalized versus the housekeeping GAPDH. * *p* < 0.05; ** *p* < 0.01; *** *p* < 0.001.

**Figure 6 ijms-24-00447-f006:**
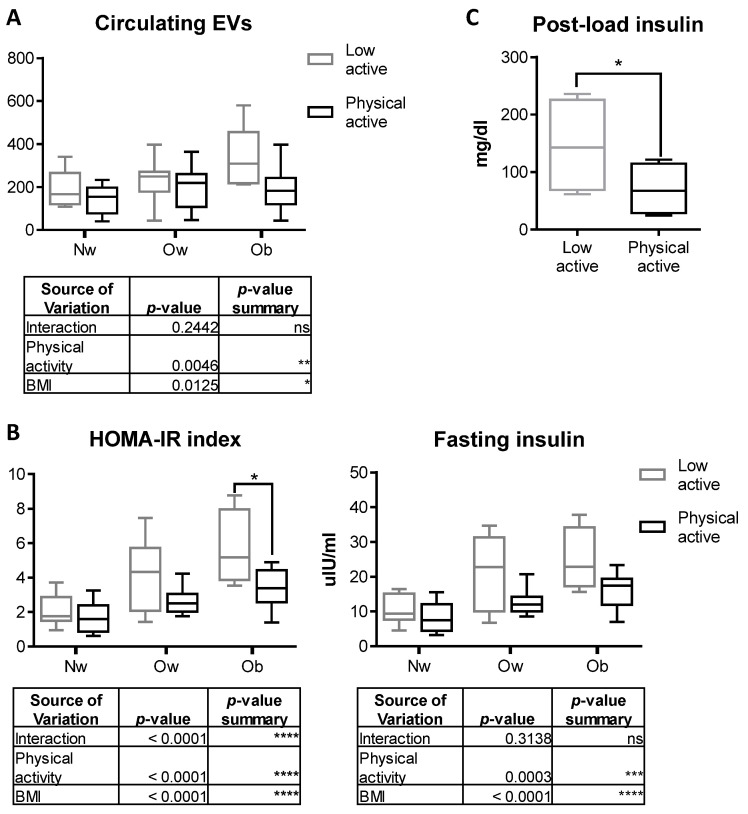
(**A**) Quantification of circulating EVs by FACS in participants with normal-weight, overweight and obesity in relation to sport activities. The table below shows the results from ANOVA 2-way analysis. (**B**) Evaluation of basal HOMA-IR Index and fasting plasma insulin in participants with normal weight, overweight and obesity performing sport activities (physically active) in comparison to their counterparts with a sedentary lifestyle (low active). (**C**) Evaluation of plasma insulin following glucose load assessed in children with obesity performing sport activities (physically active) in comparison to their counterparts with a sedentary lifestyle (low active). Normal weight group: *n* = 7 low active and *n* = 18 physically active; overweight group: *n* = 9 low active and *n* = 15 physically active; obese group: *n* = 5 low active and *n* = 15 physically active. * *p* < 0.05.

**Table 1 ijms-24-00447-t001:** Anthropometrics and biochemistry of participants.

	Normal Weight Cohort (*n* = 32)	Overweight Cohort (*n* = 28)	Obese Cohort (*n* = 22)	*p* ^1^
Sex (M/F)	13/19 (41/59%)	13/15 (46/54%)	14/8 (64/36%)	0.2
Age (years)	12.35 (10.29–18.77)	13.08 (10.73–16.17)	12.52 (9.08–16.21)	0.9
BMI ^2^ z-score (SDS)	−0.04 (−3.14–1.02)	1.43 (1.06–1.62)	2.05 (1.67–2.64)	**<0.0001**
BMI (kg/m^2^)	18.88 (12.9–25.0)	24.09 (21.3–28.7)	29.21 (22.4–38.6)	**<0.0001**
FG ^3^ (mmol/l)	83.03 (70–92)	83.29 (69–104)	89.68 (79–100)	**0.0005**
FI ^4^ (ulU/mL)	9.281 (3.2–26.0)	15.64 (6.7–34.7)	19.62 (7.0–41.7)	**<0.0001**
HOMA ^5^ -IR	1.911 (0.6–5.8)	3.192 (1.4–7.5)	4.378 (1.4–9.1)	**<0.0001**
HOMA-B	186.3 (54.0–699.4)	353.9 (100.5–2058)	267.7 (128.0–600.5)	**0.03**

^1^*p* refers to one-way ANOVA analysis, ^2^ body mass index, ^3^ fasting glucose, ^4^ fasting insulin, ^5^ homeostasis model assessment.

## Data Availability

Not applicable.

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
