# Peer review of "Circulating Extracellular Vesicles Impair Mesenchymal Stromal Cell Differentiation Favoring Adipogenic Rather than Osteogenic Differentiation in Adolescents with Obesity"

_ijms, 2022, doi:10.3390/ijms24010447_

Round 1

Reviewer 1 Report

Remarks about the manuscript: 

The authors have presented an excellent article, "Circulating extracellular vesicles impair mesenchymal stromal cell differentiation favoring adipogenic rather than osteogenic differentiation in adolescents with obesity". This manuscript focused on extracellular vesicle's crucial role in adipogenesis instead of osteogenic differentiation in obese child and adolescents. Authors have highlighted their role in impairing mesenchymal stromal cell differentiation in normal, heavyweight, and obese adolescents. They also studied comparative studies for all cases in all three child categories. Even though the sample size is small, it is sufficient evidence for the initial stage study. The data submitted in this manuscript is superb, with a proper explanation through various experimental support and analysis techniques.  

Figure 5 (A) in the manuscript needs to be appropriately aligned. 

The present manuscript contained most of the information required for publication in the IJMS journal; therefore, I will recommend it for publication. 

In my view, this manuscript (ID: IJMS-2097197) can be accepted in its current form.

Author Response

AUTHOR'S COMMENT: We thank the Reviewers for their effort in evaluating our manuscript and for their constructive comments. We have addressed all the points raised by their evaluation and performed all the changes requested, highlighted in red in the manuscript.

Reviewer 1

The authors have presented an excellent article, "Circulating extracellular vesicles impair mesenchymal stromal cell differentiation favoring adipogenic rather than osteogenic differentiation in adolescents with obesity". This manuscript focused on extracellular vesicle's crucial role in adipogenesis instead of osteogenic differentiation in obese child and adolescents. Authors have highlighted their role in impairing mesenchymal stromal cell differentiation in normal, heavyweight, and obese adolescents. They also studied comparative studies for all cases in all three child categories. Even though the sample size is small, it is sufficient evidence for the initial stage study. The data submitted in this manuscript is superb, with a proper explanation through various experimental support and analysis techniques.  

Figure 5 (A) in the manuscript needs to be appropriately aligned. 

RESPONSE: We removed the line numbering from the figure 5 that now results aligned.

The present manuscript contained most of the information required for publication in the IJMS journal; therefore, I will recommend it for publication. 

In my view, this manuscript (ID: IJMS-2097197) can be accepted in its current form.

RESPONSE: We thank the reviewer for his/her very positive evaluation.

Reviewer 2 Report

The manuscript by Barbara Peruzzi et al describes the role of circulating extracellular vesicles in adipogenic differentiation. The importance of this study is supported by the fact that obesity is turning epidemic resulting in a heavy load on the healthcare system. This work has been deisgned carefully by using an interesting approach to test the effect of EVs on cell differentiation. However, I have some comments and questions.

1.    It looks like there are overlaps between the BMI groups (e.g. normal weight is between 12.9-25 kg/m2, overweight is between 21.3-28.7, while obese 22.4-38.6), therefore it is not clear how did the authors allocate someone to a specific group?

2.    Fasting glucose concentration values seem like they are given in a different unit from what is stated, since 83 mmol/l would be extremely high.

3.    Fluorescent microscopy pictures (Figure 2 and 4) are not too convincing to support EV uptake, because Nw EV, Ow EV and Ob EV images look very similar. Has any negative and positive control (including tissue control) been used?

4.    Has any animal model been considered to monitor these effects under in vivo circumstances?

Author Response

Reviewer 2

The manuscript by Barbara Peruzzi et al describes the role of circulating extracellular vesicles in adipogenic differentiation. The importance of this study is supported by the fact that obesity is turning epidemic resulting in a heavy load on the healthcare system. This work has been deisgned carefully by using an interesting approach to test the effect of EVs on cell differentiation. However, I have some comments and questions.

  1. It looks like there are overlaps between the BMI groups (e.g. normal weight is between 12.9-25 kg/m2, overweight is between 21.3-28.7, while obese 22.4-38.6), therefore it is not clear how did the authors allocate someone to a specific group?

RESPONSE: BMI overlaps between obesity groups might have occurred since obesity status was categorised using BMI z score that also takes into account BMI differences in age and sex that are related to the accretion of body mass and the height spurt with the child growing older and not BMI as we usually do for adults. We wrongly stated the definition of each specific group in the Material and Methods section of the original version of the manuscript, and we apologize for that. We have changed the text accordingly (line 364-365 of the revised manuscript) and added a reference that explains the issue about using BMI in adolescents with obesity (reference #59).

  1. Fasting glucose concentration values seem like they are given in a different unit from what is stated, since 83 mmol/l would be extremely high.

RESPONSE: We apologize for the mistake in the unit stated for the measurement of fasting glucose concentration. The right unit is mg/dl. We changed the text (line 371 of the revised manuscript) and the Supplementary figure 4A accordingly.

  1. Fluorescent microscopy pictures (Figure 2 and 4) are not too convincing to support EV uptake, because Nw EV, Ow EV and Ob EV images look very similar. Has any negative and positive control (including tissue control) been used?

RESPONSE: In figure 2 and 4 the negative control is represented by the pictures in the columns called “Ctrl”, since no CFSE-stained EVs were added and no green fluorescence can be detected in those conditions. We did not use tissue samples as positive controls because they are not applicable in our experimental condition. Indeed, it is important to note that we are not revealing EVs into cells by immunofluorescence technique, using antibodies specific for extracellular vesicles, but we stained EVs by CFSE dye and then used stained-EVs to treat live cells, thereby monitoring dynamic EV uptake. This protocol cannot be applied to fixed tissue. As regarding the results deriving from EV uptake experiment, the quantifications showed in figure 2B and 4B demonstrate significant differences in EV uptake among different times and different EV groups, as argued in the discussion section.   

  1. Has any animal model been considered to monitor these effects under in vivo circumstances?

RESPONSE: We thank the reviewer for this interesting question. In vivo experiments have not been considered for the work presented in this manuscript, but they would be very important to definitively demonstrate our results achieved in in vitro conditions. Anyway, it is important to mention that studying EVs in in vivo conditions is a very challenge. In fact, although the in vivo injection of stained-EVs, usually by tail vein, as well as the monitoring of stained-EVs uptake by target cells could be replicated, the functional effects mediated by exogenous injected-EVs would be misrepresented by endogenous EVs-mediated effects, that cannot be prevented. Despite this, in our future works we intend to better define the differentiation status of MSC and osteoblasts isolated from genetic and not-genetic animal models of pediatric obesity, such as Ob/Ob mice or diet-induced obesity in healthy mice, respectively.